# Augmenting interpretable models with large language models during training

Chandan Singh [1] ✉, Armin Askari[2], Rich Caruana[1] & Jianfeng Gao[1]

Recent large language models (LLMs), such as ChatGPT, have demonstrated remarkable prediction performance for a growing array of tasks. However, their proliferation into high-stakes domains and compute-limited settings has created a burgeoning need for interpretability and efficiency. We address this need by proposing Aug-imodels, a framework for leveraging the knowledge learned by LLMs to build extremely efficient and interpretable prediction models. Aug-imodels use LLMs during fitting but not during inference, allowing complete transparency and often a speed/memory improvement of greater than 1000x for inference compared to LLMs. We explore two instantiations of Aug-imodels in natural-language processing: Aug-Linear, which augments a linear model with decoupled embeddings from an LLM and Aug-Tree, which augments a decision tree with LLM feature expansions. Across a variety of text-classification datasets, both outperform their non-augmented, interpretable counterparts. Aug-Linear can even outperform much larger models, e.g. a 6-billion parameter GPT-J model, despite having 10,000x fewer parameters and being fully transparent. We further explore Aug-imodels in a natural-language fMRI study, where they generate interesting interpretations from scientific data.

Large language models (LLMs) have demonstrated remarkable predictive performance across a growing range of diverse tasks[1–3]. However, their proliferation has led to two burgeoning problems. First, like most deep neural nets, LLMs have become increasingly difficult to interpret, often leading to them being characterized as black boxes and debilitating their use in high-stakes applications such as science[4], medicine[5], and policy-making[6]. Moreover, the use of black-box models such as LLMs has come under increasing scrutiny in settings where users require explanations or where models struggle with issues such as fairness[7] and regulatory pressure[8]. Second, black-box LLMs have grown to massive sizes, incurring enormous energy costs[9] and making them costly and difficult to deploy, particularly in low-compute settings (e.g., edge devices).

As an alternative to large black-box models, transparent models, such as linear models and decision trees[10] can maintain complete interpretability. Additionally, transparent models tend to be dramatically more computationally efficient than LLMs. While transparent models can sometimes perform as well as black-box LLMs[11–14], in many

settings (such as natural language processing (NLP)), there is often a considerable gap between the performance of transparent models and black-box LLMs.

We address this gap by proposing augmented-interpretable models (Aug-imodels), a framework to leverage the knowledge learned by LLMs to build extremely interpretable and efficient models. Specifically, we define an Aug-imodel as a method that leverages an LLM to fit an interpretable model but does not use the LLM during inference. This allows complete transparency and often a substantial efficiency improvement (both in terms of speed and memory). Aug-imodels can address shortcomings in existing transparent models by using the world knowledge present in modern LLMs, such as information about feature correlations.

We explore two instantiations of Aug-imodels: (i) Aug-Linear, which augments a linear model with decoupled embeddings from an LLM and (ii) Aug-Tree, which augments a decision tree with improved features generated by calling an LLM (see Fig. 1). At inference time, both are completely transparent and efficient: Aug-Linear requires

[1]Microsoft Research, Redmond, WA, USA. [2]University of California, Berkeley, Berkeley, CA, USA. ✉e-mail: chansingh@microsoft.com

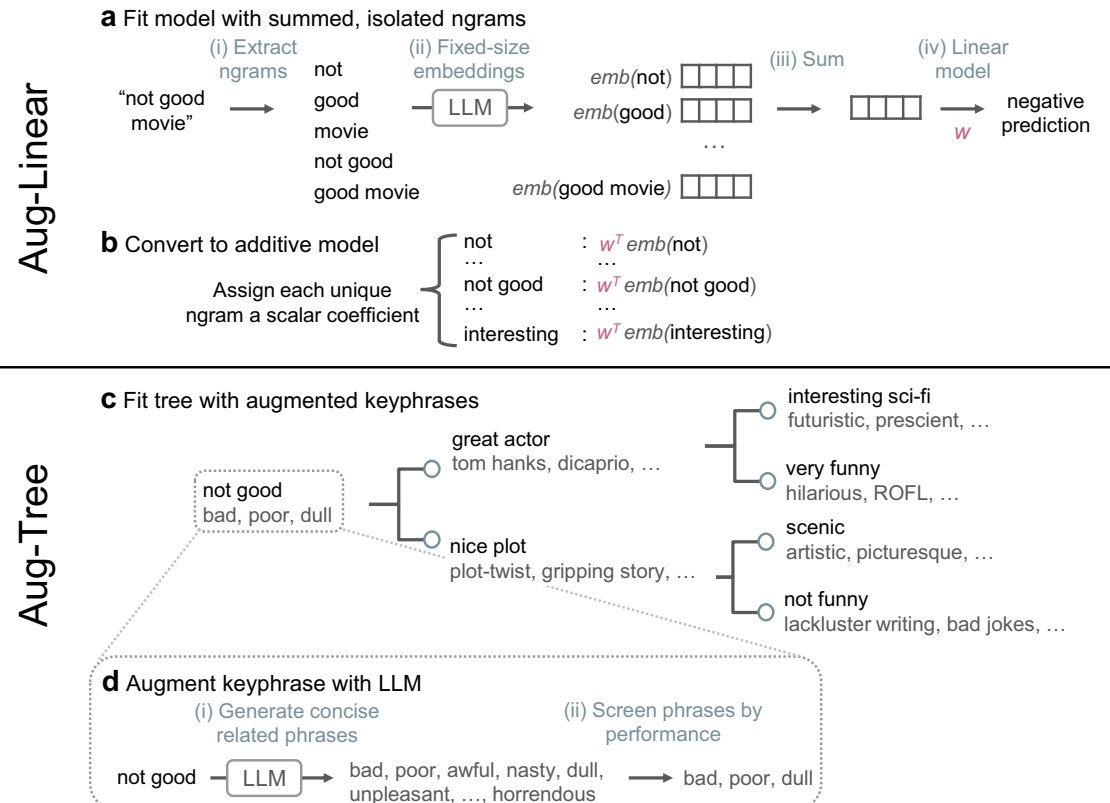

**Fig. 1 | Aug-imodels use an LLM to augment an interpretable model during fitting but not inference (toy example for movie-review classification). a** Aug-Linear fits an augmented linear model by extracting fixed-size embeddings for decoupled ngrams in a given sequence, summing them, and using them to train a supervised linear model. **b** At test-time, Aug-Linear can be interpreted exactly as a linear model. A linear coefficient for each ngram in the input is obtained by taking the dot product between the ngram's embedding and the shared vector $w$. **c** Aug-Tree improves each split of a decision tree during fitting by **d** augmenting each keyphrase found by CART with similar keyphrases generated by an LLM.

only summing coefficients from a fixed dictionary while Aug-Tree requires checking for the presence of keyphrases in an input. This allows for complete inspection of a model's decision-making process, unlike post hoc explanations, which are often unfaithful[11,15,16].

Across a variety of natural-language-processing datasets, both proposed Aug-imodels outperform their non-augmented counter-parts. Aug-Linear can even outperform much larger models, (e.g., a 6-billion parameter Generative pretrained transformer model, GPT-J[17]), despite having 10,000x fewer parameters and no nonlinearities. We further explore Aug-imodels in a natural-language fMRI context, where we find that they can predict well and generate interesting interpretations. In what follows, the section "Results" shows results for predictive performance and interpretation, the section "Discussion"

includes a discussion, and the section "Methods" formally introduces Aug-imodels.

## Results

### Experimental setup for evaluating text-classification performance

Table 1 shows the datasets we study: four widely used text-classification datasets spanning different domains (e.g., classifying the emotion of tweets[18], the sentiment of financial news sentences[19], or the sentiment of movie reviews[20, 21]), and one scientific text regression dataset (described in section "fMRI results: analyzing fMRI data with Aug-imodels")[22]. Across datasets, the number of unique ngrams grows quickly from unigrams to bigrams to trigrams. Moreover, many ngrams appear very rarely, e.g., in the Financial Phrasebank (FPB) dataset, 91% of trigrams appear only once in the training dataset.

We compare Aug-Linear to four interpretable baseline models: Bag of ngrams, TF-IDF (Term frequency-inverse document frequency)[23], GloVE[24] (we use the pre-trained Glove embeddings trained on Common Crawl containing 840 billion tokens, 2.2 million vocab-size, cased, 300-dimensional vectors), and a model trained on BERT embeddings for each unigram in the input (which can be viewed as running Aug-Linear with only unigrams). We use BERT (`bert-base-uncased`)[3] as the LLM for extracting embeddings, after finetuning on each dataset; see Supplementary Table 1 for details on all models and downloadable checkpoints. In each case, a model is fit via cross-validation on the training set (to tune the amount of $\ell_2$ regularization added) and its accuracy is evaluated on the test set.

We compare Aug-Tree to two decision-tree baselines: CART[10] and ID3[25], and we use bigram features. In addition to individual trees, we fit

**Table 1 | Overview of datasets studied here**

|  | FPB | Rotten tomatoes | SST2 | Emotion | fMRI |
|---|---|---|---|---|---|
| Samples (train) | 2313 | 8530 | 67,349 | 16,000 | 9461 |
| Samples (val) | 1140 | 1066 | 872 | 2000 | 291 |
| Classes | 3 | 2 | 2 | 6 | Regression |
| Unigrams | 7169 | 16,631 | 13,887 | 15,165 | 4980 |
| Bigrams | 28,481 | 93,921 | 72,501 | 106,201 | 27,247 |
| Trigrams | 39,597 | 147,426 | 108,800 | 201,404 | 46,834 |
| Trigrams that appear only once | 91% | 93% | 13% | 89% | 71% |

The number of ngrams grows quickly with the size of the ngram.

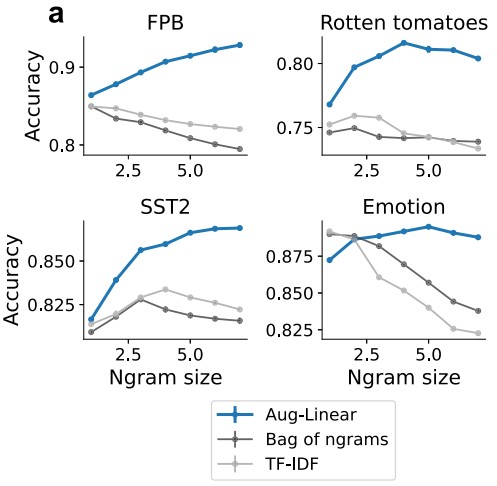

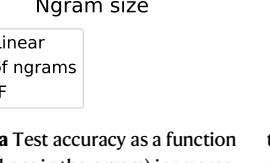

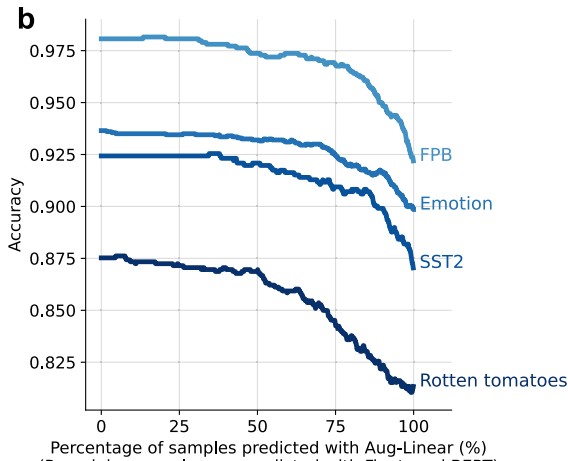

**Fig. 2 | Text-classification accuracy for Aug-Linear. a** Test accuracy as a function of ngram size. As the ngram size (i.e., the number of tokens in the ngram) increases, the gap between Aug-Linear and the baselines grows. Averaged over three random cross-validation splits; error bars are standard errors of the mean (many are within the points). **b** Accuracy when predicting using a 2-step procedure: uses Aug-Linear predictions on samples for which Aug-Linear is confident and Finetuned BERT predictions on the remaining samples. A large percentage of samples can be accurately predicted with Aug-Linear without a significant drop in performance.

## Table 2 | Test accuracy for different models

| | | FPB | Rotten tomatoes | SST2 | Emotion | AVG |
|---|---|---|---|---|---|---|
| Ours | Aug-Linear | 92.8 ± 0.37 | 81.6 ± 0.05 | 86.9 ± 0.10 | 89.5 ± 0.03 | 87.7 |
| Interpretable baselines | Bag of ngrams | 85.0 ± 0.11 | 75.0 ± 0.09 | 82.8 ± 0.00 | 89.0 ± 0.09 | 83.0 |
| | TF-IDF | 84.9 ± 0.16 | 75.9 ± 0.06 | 83.4 ± 0.11 | 89.2 ± 0.04 | 83.4 |
| | GloVe | 80.5 ± 0.06 | 78.7 ± 0.03 | 80.1 ± 0.10 | 73.1 ± 0.09 | 78.1 |
| | BERT unigram embeddings | 86.4 ± 0.13 | 76.8 ± 0.19 | 81.7 ± 0.07 | 87.2 ± 0.06 | 83.0 |
| Black-box baselines | BERT finetuned | 98.0 | 87.5 | 92.4 | 93.6 | 92.9 |
| | GPT-3 | 39.6 ± 1.6 | 82.7 ± 3.3 | 90.5 ± 3.9 | 45.1 ± 4.1 | 64.5 |
| | GPT-J | 27.0 ± 1.9 | 58.9 ± 3.1 | 58.4 ± 2.8 | 19.3 ± 1.9 | 40.9 |

Aug-Linear yields improvements over interpretable baselines and is competitive with some black-box baselines. See results for more datasets in Supplementary Table 3. Errors show standard error of the mean over 3 random data splits (or 3 different prompts for GPT models).

bagging ensembles, where each tree is created using a bootstrap sample the same size as the original dataset (as done in Random Forest[26]) and has depth 8. This hurts interpretability but can improve predictive performance and calibration. For simplicity, we run Aug-Linear only in a binary classification setting; to do so, we take two opposite classes from each multi-class dataset (*Negative/Positive* for *FPB* and *Sadness/Joy* for *Emotion*).

### Aug-Linear text-classification performance

Figure 2a shows the test accuracy of Aug-Linear as a function of the ngram size used for computing features. Aug-Linear outperforms the interpretable baselines, achieving a considerable increase in accuracy across three of the four datasets. Notably, Aug-Linear accuracy increases with ngram size, whereas the accuracy of baseline methods decreases or plateaus. This is likely due to Aug-Linear fitting only a fixed-size parameter vector, helping to prevent overfitting.

Table 2 shows the test accuracy results for various models when choosing the size of ngrams via cross-validation. Compared with interpretable baselines, Aug-Linear shows considerable gains on three of the datasets and only a minor gain on the tweet dataset (*Emotion*), likely because this dataset requires fitting less high-order interactions.

Compared with the zero-shot performance of the much larger GPT models (6-billion parameter GPT-J[17] and 175-billion parameter GPT-3, `text-davinci-002`[1]). Accuracy for GPT models is computed by averaging over human-written prompts taken from PromptSource[27]; see details in Supplementary section 1). Aug-Linear

outperforms GPT-J. Aug-Linear lags slightly behind GPT-3 for binary classification problems (*Rotten Tomatoes* and *SST2*) but outperforms GPT-3 for multi-class classification problems (*FPB* and *Emotion*). The best black-box baseline (a BERT finetuned model) outperforms Aug-Linear by 4%–6% accuracy. This is potentially a reasonable tradeoff in settings where interpretability, speed, or memory bottlenecks are critical.

At inference time, it may be useful to use Aug-Linear on relatively easy samples (for interpretability/memory/speed/cost-saving) but relegate difficult samples to a black-box model. To study this setting, we predict each sample with a 2-step procedure. First, we predict the sample with Aug-Linear. If its prediction confidence is high (the predicted probability for the top class is above some threshold), we return the Aug-Linear prediction. Otherwise, we predict the sample using the black-box model. If Aug-Linear is well-calibrated, it should perform well in this setting, since it can relegate the samples where it performs poorly to the black-box model (here, we use Finetuned BERT as the black-box model).

Figure 2b shows the accuracy of the entire test set in this setting. We vary the confidence threshold that decides whether to predict using Aug-Linear or Finetuned BERT; this results in a curve showing accuracy as a function of the percentage of samples predicted with Aug-Linear. Since Aug-Linear predictions are well-calibrated (see Supplementary Fig. 1), rather than the accuracy linearly interpolating between Aug-Linear and BERT, a large percentage of samples can be predicted with Aug-Linear while incurring little to no drop in accuracy.

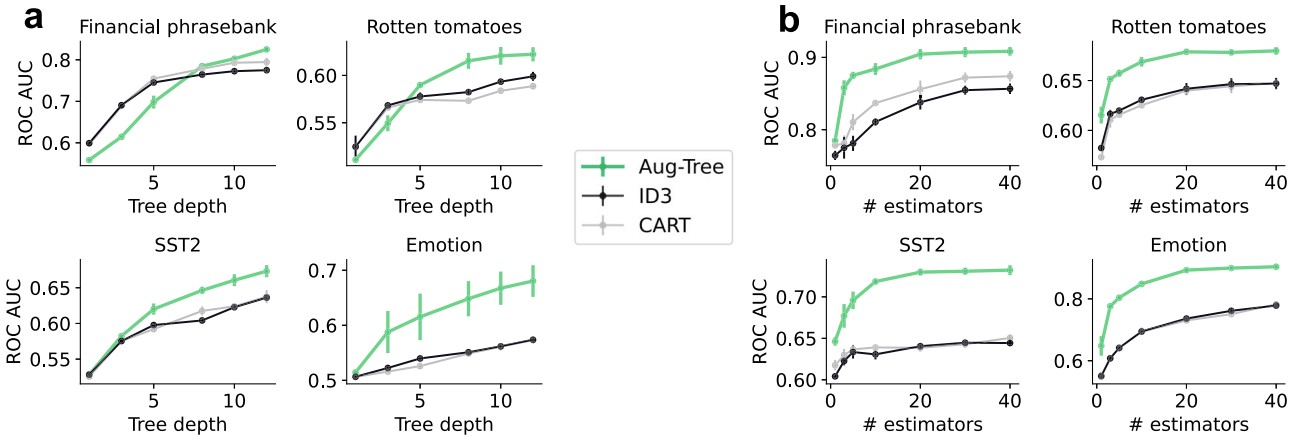

**Fig. 3 | Aug-Tree text-classification performance.** Test performance as a function of **a** tree depth for individual trees and **b** number of estimators in a bagging ensemble. Values are averaged over 3 random dataset splits; error bars show the standard error of the mean (many are within the points).

For example, when using Aug-Linear on 50% of samples, the average drop in test accuracy is only 0.0053.

In cases involving inference memory/speed, Aug-Linear can be converted to a dictionary of coefficients, whose size is the number of ngrams that appeared in training (see Table 1). For a trigram model, this yields roughly a 1000x reduction in model size compared to the ~110 million trainable parameters in BERT, with much room for further size reduction (e.g., simply removing coefficients for trigrams that appear only once yields another 10-fold size reduction). Inference is nearly instantaneous, as it requires looking up coefficients in a dictionary and then a single sum (and does not require a GPU).

Supplementary section 1.1 explores accuracy/efficiency tradeoffs. For example, Aug-Linear performance degrades gracefully when the model is compressed by removing its smallest coefficients. In fact, the test accuracy of Aug-Linear models trained with 4-grams on the *Emotion* and *Financial phrasebank* datasets actually improves after removing 50% of the original coefficients (Supplementary Fig. 2A). Additionally, one can vary the size of ngrams used at test-time without severe performance drop, potentially enabling compressing the model by orders of magnitude (see Supplementary Figs. 2B and 3). For example, when fitting a model with 4-grams and testing with 3-grams, the average performance drop is ~2%.

Supplementary Table 2 shows how generalization accuracy changes when the LLM used to extract embeddings for Aug-Linear is varied (e.g., using GPT-2, RoBERTA, or LlaMa), or different layers/ngram selection techniques are used. Supplementary Table 3 shows results for more multi-class datasets and when varying tokenization schemes. Across the variations, embeddings from finetuned models and larger models tend to yield better results.

## Aug-Tree text-classification performance

We now investigate the predictive performance of Aug-Tree, measured by the test ROC AUC on the previous text-classification datasets altered for binary classification. Note that the performance of all tree-based methods on the studied datasets is below the performance of the GLM methods in the section "Aug-Linear text-classification performance" (see Supplementary Table 7 for a direct comparison). Nevertheless, Aug-Tree models maintain potential advantages, such as storing far fewer parameters, clustering important features together, and better modeling long-range interactions.

Figure 3a shows the performance of Aug-Tree as a function of tree depth compared to decision-tree baselines. Aug-Tree shows improvements that are sometimes small (e.g., for *Financial phrasebank*) and sometimes relatively large (e.g., for *Emotion*). Figure 3b shows the performance of a bagging ensemble of trees with different tree methods used as the base estimator. Here, using Aug-Tree shows a

reliable and significant gain across all datasets compared to ensembles of baseline decision-tree methods. This suggests that LLM augmentation may help to diversify or decorrelate individual trees in the ensemble. Supplementary Table 6 shows variations of different hyperparameters for Aug-Tree, such as using embeddings or dataset-specific prompts to expand keyphrases.

## Interpretation results: interpreting fitted models

In this section, we interpret fitted Aug-imodels. We first inspect an Aug-Linear model fitted using unigram and bigram features on the *SST2* dataset which achieves 84% test accuracy. Next, we analyze the keyphrase expansions made in fitted Aug-Tree bagging ensembles.

A fitted Aug-Linear model can be interpreted for a single prediction (i.e., getting a score for each ngram in a single input, as in Fig. 1) or for an entire dataset (i.e., by inspecting its fitted coefficients). Figure 4a visualizes the fitted Aug-Linear coefficients (i.e., the contribution to the prediction $w^T \phi(x_i)$) with the largest absolute values across the SST2 dataset. To show a diversity of ngrams, we show every fifth ngram. The fitted coefficients are semantically reasonable and many contain strong interactions (e.g., *not very* is assigned to be negative whereas *without resorting* is assigned to be positive). This form of model visualization allows a user to audit the model with prior knowledge. Note that the coefficient for an ngram, e.g., *not bad* (positive) is not simply the sum of its constituent ngrams: *not* (negative) and *bad* (negative), see Supplementary Fig 5. Moreover, these coefficients are exact and therefore avoid summarizing interactions, making them considerably more faithful than post hoc methods, such as LIME[28] and SHAP[29] (see Supplementary section 1.2 for a comparison).

Figure 4b compares the fitted Aug-Linear coefficients to human-labeled sentiment phrase scores for unigrams/bigrams in SST (note: these continuous scores are separate from the binary sentence labels used for training in the SST2 dataset). Both are centered, so that 0 is neutral sentiment and positive/negative values correspond to positive/negative sentiment, respectively. There is a strong positive correlation between the coefficients and the human-labeled scores (Spearman rank correlation $\rho = 0.63$), which considerably improves over coefficients from a bag-of-bigrams model trained on SST ($\rho = 0.39$).

One strength of Aug-Linear is its ability to infer linear coefficients for ngrams that were not seen during training. Whereas baseline models generally assign each unknown ngram the same coefficient (e.g., 0), Aug-Linear can effectively assign these new ngrams accurate coefficients. As one example, Fig. 4c shows that the Aug-Linear model trained only on bigrams in Fig. 4a, b can automatically infer coefficients for trigrams (which were not fit during training). The inferred coefficients are semantically meaningful, even capturing three-way interactions, such as *not very amusing*. To show a diversity of ngrams, we show

every 20th ngram. Figure 4d shows the coefficients compared to the human-labeled SST phrase sentiment for all trigrams in SST. Again, there is a strong correlation, where the Aug-Linear coefficients achieve a rank correlation $\rho = 0.71$, which even outperforms the bag-of-words model directly trained on trigrams ($\rho = 0.49$).

A fitted Aug-Tree model can be easily interpreted for a single prediction (i.e., by inspecting the ngrams that triggered relevant splits) or by visualizing the entire tree (e.g., Fig. 1c). Here, we additionally analyze how well each ngram found by CART matches the augmented ngrams found by the LLM; the better this match is, the easier it is to interpret a split.

Table 3 shows examples of the ngrams that were most frequently augmented when fitting a bagging ensemble of 40 Aug-Tree s to the four text-classification datasets in Table 1. Added ngrams seem qualitatively reasonable, e.g., the keyphrase *good* expands to *fine, highly, solid,…, valuable*. We evaluate how well the expansions match the original CART ngram via human evaluation scores. Human evaluators are given the original ngram and the added ngrams, then instructed "You are given a keyphrase along with related keyphrases. On a scale of

1 (worst) to 5 (best), how well do the related keyphrases match the example keyphrase?" Human evaluation scores are averaged over 3 Ph.D. students in machine learning not affiliated with the study and 15 random ngrams from each dataset. Table 3 shows that the average human score for splits in each dataset is consistently greater than 4. This is substantially higher than the baseline score of 1.3 assigned by human evaluators when 15 ngrams and expansions are randomly paired and evaluated. Supplementary Table 5 gives more details on ngram expansions.

## fMRI Results: analyzing fMRI data with Aug-imodels

We now explore Aug-imodels in a real-world neuroscience context. A central challenge in neuroscience is understanding how and where semantic concepts are represented in the brain. To meet this challenge, one line of study predicts the response of different brain voxels (i.e., small regions in space) to natural-language stimuli. We analyze data from a recent study in which the authors collect functional MRI (fMRI) responses as human subjects listen to hours of narrative stories[22]. The fMRI responses studied here contain 95,556 voxels from

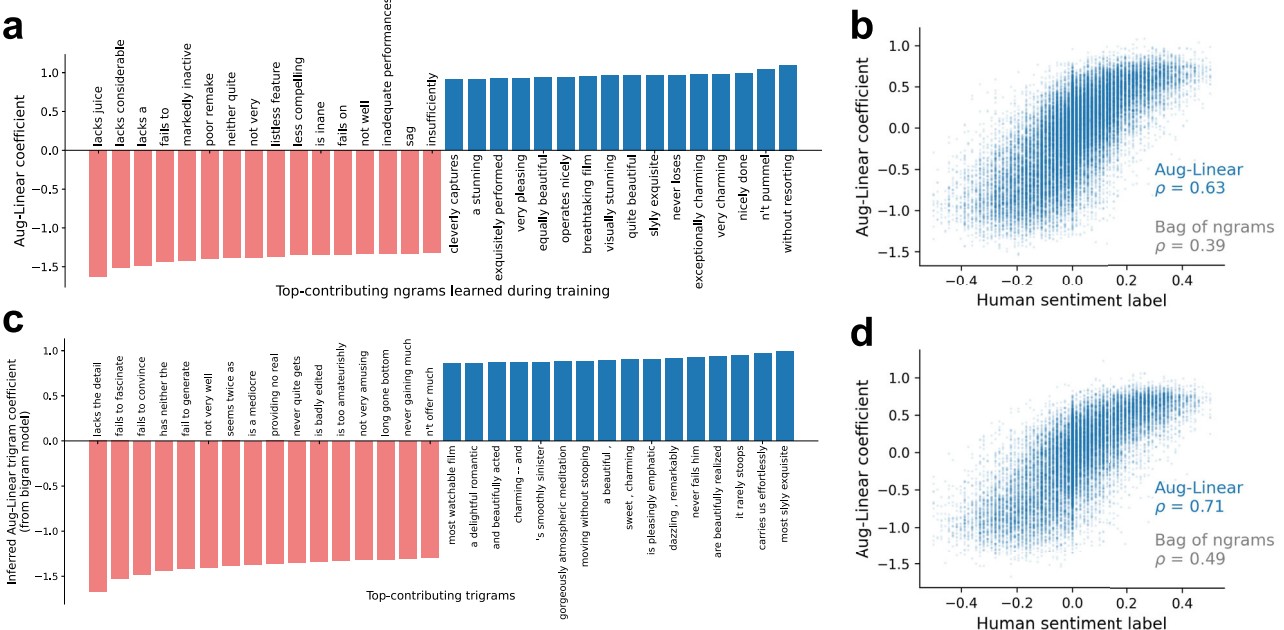

**Fig. 4 | Interpreting Aug-Linear.** Top and bottom contributing ngrams to an Aug-Linear model trained on SST2 bigrams are **a** qualitatively semantically accurate and **b** match human-labeled phrase sentiment scores. For the same Aug-Linear model, which is trained only on bigrams, inferred trigrams coefficients are **c** qualitatively semantically accurate and **d** match human-labeled phrase sentiment scores.

**Table 3 | Examples of most frequently augmented ngrams for each dataset when fitting an ensemble of 40 Aug-Tree**

| Dataset | Human score | Example CART ngram | Added ngrams |
|---|---|---|---|
| SST2 | 4.6 ± 0.1 | good | fine, highly, solid, worthy, pleasing, satisfactory, outstanding, honorable, unwavering, valuable,… |
| | | best | most remarkable, outstanding, superb, flawless, splendid, superlative, exceptional, impeccable,… |
| RT | 4.4 ± 0.1 | dull | dreary, uninteresting, lackluster, listless, lifeless, uninspired, wearisome, drab, joylessly,… |
| | | bad | unpleasant, dire, despicable, terrible, heinous, disgusting, vile, putrid, atrocious, nasty, poor,… |
| Emotion | 4.4 ± 0.2 | miserable | gloomy, disillusioned, pathetic, doomed, agonized, despairing, pointless, despondent,… |
| | | sorry | embarrassed, sorrowful, remorseful, excuse, unsatisfied, guilt, regretful, forgive, apologies,… |
| FPB | 4.2 ± 0.2 | increased | widened, consolidated |
| | | fell | slipped, slumped, diminished, plunged, dropped, weakened, lost ground |

Human scores measure the similarity between an ngram and its expansion. They range from 1 (worst match) to 5 (best match), and the baseline score when ngrams and expansions are randomly paired and evaluated is 1.3 ± 0.1. Error bars show the standard error of the mean.
*FPB* Financial Phrasebank, *RT* rotten tomatoes.

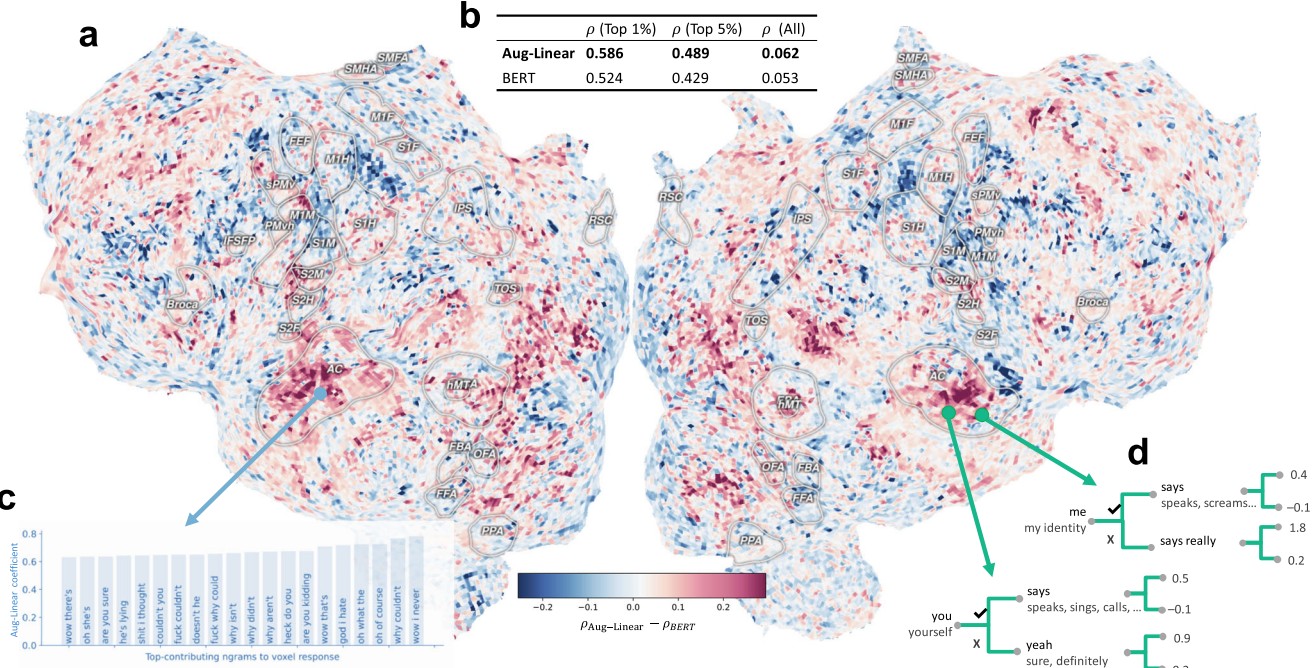

| | $\rho$ (Top 1%) | $\rho$ (Top 5%) | $\rho$ (All) |
|---|---|---|---|
| **Aug-Linear** | **0.586** | **0.489** | **0.062** |
| BERT | 0.524 | 0.429 | 0.053 |

**Fig. 5 | Aug-imodels prediction performance and interpretation for fMRI voxels. a** Map of the difference between the performance of Aug-Linear and BERT for fMRI voxel prediction across the cortex. Positive values (red) show where Aug-Linear outperforms BERT, measured by correlation on the test set. **b** Aug-Linear outperforms BERT when averaging across all voxels, or just over the 1%/5% with the highest test correlations. Standard errors of the mean are all less than 0.0015. **c** Example Aug-Linear model for a single voxel, visualized with the top Aug-Linear coefficients. **d** Example Aug-Tree model for two voxels.

a single subject, with 9461 time points used for training/cross-validation and 291 time points used for testing. We predict the continuous response for each voxel at each time point using the 20 words that precede the time point. We skip the most recent 4 words due to account for a time delay in the fMRI BOLD response. Seminal work on this task found that linear models of word vectors could effectively predict voxel responses[30], and more recent work shows that LLMs can further improve predictive performance[31, 32]. Aug-Linear is well-suited to this task, as it combines low-level word information with the contextualized information present in higher-order ngrams, both of which have been found to contribute to fMRI representations of text[33].

Figure 5a visualizes the voxels in the cortex which are better predicted by Aug-Linear than BERT. The improvements are often spatially localized within well-studied brain regions such as the auditory cortex (AC). Figure 5b shows that the test performance for Aug-Linear (measured by the Pearson correlation coefficient $\rho$) outperforms the black-box BERT baseline. Supplementary section 3 gives further data details and comparisons, e.g., Aug-Linear also outperforms other linear baselines.

Going beyond prediction performance, Fig. 5c investigates a simple example of how Aug-Linear could help interpret an underlying brain region. We first select the voxel which is best predicted by Aug-Linear (achieving a test correlation of 0.76) and then visualize the largest fitted Aug-Linear coefficients for that voxel. These correspond to which ngrams increase the activity of the fMRI voxel the most. Interestingly, these ngrams qualitatively correspond to understandable concepts: *questioning*, e.g., "are you sure", often combined with *disbelief/incredulity*, e.g. "wow I never". Figure 5d shows two examples of voxels that are better predicted by Aug-Tree than Aug-Linear (Aug-Tree yields test correlations of 0.35 and 0.36). These two voxels are both related to someone speaking, but they seem to depend on interactions between the noun (*me* or *you*) and the verb (*says*). To elicit a large response both must be present, something which is difficult to capture in additive models, even with ngrams, since these words may not be close together in a sentence.

This interpretation approach could be applied more rigorously to generate hypotheses for text inputs that activate brain regions, and then test them with follow-up fMRI experiments.

## Discussion

Aug-imodels provide a promising direction towards future methods that reap the benefits of both LLMs and transparent models in NLP: high accuracy along with interpretability/efficiency. This potentially opens the door for introducing LLM-augmented models in high-stakes domains, such as medical decision-making and in new applications on compute-limited hardware. Aug-imodels are currently limited to applications for which an effective LLM is available, and thus may not work well for very esoteric NLP tasks. However, as LLMs improve, the predictive performance of Aug-imodels should continue to improve and expand to more diverse NLP tasks. More generally, Aug-imodels can be applied to domains outside of NLP where effective foundation models are available (e.g., computer vision or protein engineering).

Though helpful, Aug-imodels are limited by their transparent model form and cannot capture some complex interactions that LLMs can model. To remedy this, Aug-imodels could be readily extended beyond linear models and trees to improve transparent models such as rule lists, prototype-based models, symbolic models, and rule sets with LLM augmentation during training time. In all these cases, LLM augmentation could use LLM embeddings (as is done in Aug-Linear), use LLM generations (as is done in Aug-Tree), or use LLMs in new ways. Aug-Linear could be extended to nonlinearly transform the embedding for each ngram with a model before summing to obtain the final prediction, similar to the nonlinearity present in generalized additive models (GAMs) such as the explainable boosting machine[34],. Additionally, Aug-Linear could fit long-range interaction terms as opposed to only ngrams. Aug-Tree could leverage domain knowledge to engineer more meaningful prompts for expanding ngrams or for extracting relevant ngrams. Both models can be further studied to improve their compression (potentially with LLM-guided compression techniques) or to extend their capabilities to tasks beyond classification/

regression, such as sequence prediction or outlier detection. We hope that the introduction of Aug-imodels can help push improved performance prediction into high-stakes applications, improve interpretability for scientific data, and reduce unnecessary energy/compute usage.

## Methods

In this section, the section "Limitations of existing transparent methods" overviews the limitations of existing transparent methods, section "Aug-Linear method description" introduces Aug-Linear, and the section "Aug-Tree method description" introduces Aug-Tree.

### Limitations of existing transparent methods

We are given a dataset of $n$ natural-language strings $X_{text}$ and corresponding labels $\mathbf{y} \in \mathbb{R}^n$. In transparent modeling, often each string $x$ is represented by a bag-of-words, in which each feature $x_i$ is a binary indicator (or count) of the presence of a single token (e.g., the word *good*). To model interactions between tokens, one can instead use a bag-of-ngrams representation, whereby each feature is formed by concatenating multiple tokens (e.g., the phrase *not good*). Using a bag-of-ngrams representation maps $X_{text}$ to a feature matrix $X \in \mathbb{R}^{n \times p}$, where $p$ is the number of unique ngrams in $X_{text}$. While this representation enables interpretability, the number of ngrams in a dataset grows exponentially with the size of the ngram (how many tokens it contains) and the vocab-size; even for a modest vocab-size of 10,000 tokens, the number of possible trigrams is already $10^{12}$. This makes it difficult for existing transparent methods to model all trigrams without overfitting. Moreover, existing transparent methods completely fail to learn about ngrams not seen in the training set.

**Preliminaries: linear models.** We build on generalized linear models, or GLMs[35], which take the form:

$$g(\mathbb{E}[y]) = \beta_0 + \sum_{i=1}^{p} \beta_i \cdot x_i \qquad (1)$$

where $(x_1, x_2, \ldots, x_p)$ are the input features (i.e., ngrams), $y$ is the target variable, $g(\cdot)$ is the link function (e.g., logistic function) and each $\beta_i$ is a scalar coefficient. Due to the function's additivity, the contribution of each feature can be interpreted independently.

**Preliminaries: decision trees.** CART[10] fits a binary decision tree via recursive partitioning. When growing a tree, CART chooses for each node $t$ the split $s$ that maximizes the impurity decrease in the responses $\mathbf{y}$. For a given node $t$, the impurity decrease has the expression

$$\hat{\Delta}(s, t, \mathbf{y}) := \sum_{\mathbf{x}_i \in t} h(y_i, \bar{y}_t) - \sum_{\mathbf{x}_i \in t_L} h(y_i, \bar{y}_{t_L}) - \sum_{\mathbf{x}_i \in t_R} h(y_i, \bar{y}_{t_R}), \qquad (2)$$

where $t_L$ and $t_R$ denote the left and right child nodes of $t$ respectively, and $\bar{y}_t, \bar{y}_{t_L}, \bar{y}_{t_R}$ denote the mean responses in each of the nodes. For classification, $h(\cdot, \cdot)$ corresponds to the Gini impurity, and for regression, $h(\cdot, \cdot)$ is the mean-squared error. Each split $s$ is a partition of the data based on a feature in $X$. To grow the tree, the splitting process is repeated recursively for each child node until a stopping criteria (e.g., a max depth) is satisfied. At inference time, we predict the response of an example by following its path from the root to a leaf and then predicting with the mean value found in that leaf.

### Aug-Linear method description

To remedy the issues with the GLM model in Eq. (1), we propose Aug-Linear, an intuitive model which leverages a pre-trained LLM to extract a feature representation $\phi(x_i)$ for each input ngram $x_i$. This allows learning only a single linear weight vector $w$ with a fixed dimension (which depends on the embedding dimension produced by the LLM), regardless of the number of ngrams. As a result, Aug-Linear can learn efficiently as the number of input features grows, and can also infer coefficients for unseen features. The fitted model is still a GLM, ensuring that the model can be precisely interpreted as a linear function of its inputs:

$$g(\mathbb{E}[y]) = \beta + w^T \sum_i \phi(x_i) \qquad (3)$$

Fitting Aug-Linear resembles learning a linear layer on top of word embeddings[24,36,37], but instead uses LLM ngram embeddings to better compare the semantics/interactions present within an ngram. Aug-Linear is also similar to the approach of finetuning a single linear layer on top of LLM embeddings[38], but instead separately extracts/embeds each ngram to keep the contributions to the prediction strictly additive across ngrams (see Fig. 1a):

(i)   *Extracting ngrams*: To ensure input ngrams are interpretable, ngrams are constructed using a word-level tokenizer (here, spaCy[39]). We select the size of ngrams to be used via cross-validation.

(ii)  *Extracting embeddings*: Each ngram is separately fed through the LLM to retrieve a fixed-size embedding. When feeding each ngram through, we apply the standard preprocessing and tokenizer used by the LLM. For example, when the LLM is BERT[3], we prepend `[CLS]` to the ngram, append `[SEP]` to it, and use BERT's word-piece tokenizer to process the resulting string into tokens (note that this splits an ngram into many tokens). We then average over the dimension corresponding to the number of tokens to yield a fixed-size embedding (a common alternative for bi-directional (masked) language models is to use the embedding for a special token, i.e., `[CLS]`, but we aim to keep the approach here more general).

(iii) *Summing embeddings*: The embeddings of each ngram in the input are summed to yield a single fixed-size vector, ensuring additivity of the final model.

(iv)  *Fitting the final linear model to make predictions*: A linear model is fit on the summed embedding vector. We choose the link function $g$ to be the logit function (or the softmax for multi-class) for classification and the identity function for regression. In both cases, we add $\ell_2$ regularization over the parameters $w$ in Eq. (3) and minimize the loss (cross-entropy for classification, mean-squared error for regression) using Limited memory BFGS (optimization is performed using scikit-learn[40]).

**Computational considerations.** During fitting, Aug-Linear is inexpensive to fit as (1) the pre-trained LLM is not modified in any way, and can be any existing off-the-shelf model and (2) Aug-Linear only requires fitting a fixed-size linear model. After training, the model can be converted to a dictionary of scalar coefficients for each ngram, where the coefficient is the dot product between the ngram's embedding and the fitted weight vector $w$ (Fig. 1b). This makes inference extremely fast and converts the model to have size equal to the number of fitted ngrams. When new ngrams are encountered at testtime, the coefficients for these ngrams can optionally be inferred by again taking the dot product between the ngram's embedding and the fitted weight vector $w$.

### Aug-Tree method description

Aug-Tree improves upon a CART decision tree by augmenting features with generations from an LLM. This helps capture correlations between ngrams, including correlations with ngrams that are not present in the training data. Aug-Tree does not modify the objective in Eq. (2) but rather modifies the procedure for fitting each individual split $s$ (Fig. 1d). While CART restricts each split to a single ngram, Aug-

Tree lets each split fit a disjunction of ngrams (e.g., *ngram1* ∧ *ngram2* ∧ *ngram3*). The disjunction allows a split to capture sparse interactions, such as synonyms in natural language. This can help mitigate overfitting by allowing individual splits to capture concrete concepts, rather than requiring many interacting splits.

When fitting each split, Aug-Tree starts with the ngram which maximizes the objective in Eq. (2), just as CART would do, e.g., *not good*. Then, we query an LLM to generate similar ngrams to include in the split, e.g., *bad, poor, awful,..., horrendous*. Specifically, we query GPT-3 (`text-davinci-003`)[1] with the prompt *Generate 100 concise phrases that are very similar to the keyphrase:\nKeyphrase: "{keyphrase}"\n1.* and parse the outputs into a list of ngrams. We greedily screen each ngram by evaluating the impurity of the split when including the ngram in the disjunction; we then exclude any ngram that increases the split's impurity, resulting in a shortened list of ngrams, e.g., *bad, poor, dull*.

**Computational considerations.** As opposed to Aug-Linear, Aug-Tree uses an LLM API rather than LLM embeddings, which may be more desirable depending on access to compute. The number of LLM calls required is proportional to the number of nodes in the tree. During inference, the LLM is no longer needed, and making a prediction simply requires checking an input for the presence of specific ngrams along one path in the tree. Storing an Aug-Linear model requires memory proportional to the number of raw strings associated with tree splits, usually substantially reducing memory over the already small Aug-Linear model. We explore variations of Aug-Tree (such as using LLM embeddings rather than an API) in Supplementary section 2.

### Background and related work

**Improving linear models with neural networks.** There is a large literature on additive models being used for interpretable modeling. This includes GAMs[41], which have achieved strong performance in various domains by modeling individual component functions/interactions using regularized boosted decision trees[34] and more recently using neural networks[42]. However, existing GAM methods are limited in their ability to model the high-order feature interactions that arise in NLP. Meanwhile, NLP has seen great success in models which build strong word-level representations, e.g., word2vec[36,37], GloVe[24], FastText[43], and ELMo[44]. By replacing such models with LLM embeddings, Aug-Linear enables easily modeling ngrams of different lengths without training a new model. Moreover, unlike earlier methods, LLMs can incorporate information about labels into the embeddings (e.g., by first finetuning an LLM on a downstream prediction task).

**Decision trees.** There is a long history of greedy methods for fitting decision trees, e.g., CART[10] or ID3[25]. More recent work has explored fitting trees via global optimization rather than greedy algorithms[45–47]; this can improve performance given a fixed tree size but incurs a high computational cost. Other recent studies have improved trees after fitting through regularization[48] or iterative updates[49]. Some recent works have studied using trees as a way to guide large language models[50,51]. Beyond trees, there are many popular classes of rule-based models, such as rule sets[52], rule lists[53,54], and tree sums[14]. Aug-Tree addresses a common problem shared by rule-based approaches: modeling the sparse, correlated features that are common in tasks such as text classification.

Beyond fitting a single tree, tree ensembles such as Random Forest[26], gradient-boosted trees[55], XGBoost[56], and BART[57], have all shown strong predictive performance in diverse settings. These ensembling approaches are compatible with Aug-Tree, as they can be used as the base estimator in any of these approaches.

**Interpreting features and feature interactions.** Related to this work is post hoc methods that aim to help understand a black-box model, i.e.,

by providing feature importances using methods such as LIME[28], SHAP[58], and others[59,60]. Slightly more related are works that aim to explain feature interactions or transformations in neural networks[61–63] However, all these methods lose some information by summarizing the model and suffer from issues with summarizing interactions[64,65]. Alternative forms of explanation exist specifically for NLP, such as extractive rationales[66,67], natural-language explanations for individual predictions[68,69], and more recently LLM-generated explanations (e.g., a chain of thought[70]). All these methods fail to explain the model *as a whole* and are again less reliable than having a fully transparent model (e.g., explanations are often unfaithful[15,16]).

**Interpreting/distilling neural networks.** Alternatively, one can investigate whether an LLM's learned representations via probing[71,72] or by directly analyzing a model's internal weights and activations[73–75]. The work here is related to studies that aim to make neural networks more interpretable. For example, models can make predictions by comparing inputs to prototypes[76,77], by predicting intermediate interpretable concepts[78–80], using LLMs to extract prompt-based features[81,82], distilling a neural network into a mostly transparent model[83,84] or distilling into a fully transparent model (e.g., adaptive wavelets[12] or an additive model[85]). Separately, many works use neural network distillation to build more efficient (but still black-box) neural network models, e.g., refs. [86,87].

### Reporting summary

Further information on research design is available in the Nature Portfolio Reporting Summary linked to this article.

## Data availability

All data is available open-source and instructions for downloading the data are available at github.com/microsoft/augmented-interpretable-models. Text-classification datasets can be downloaded from huggingface using the huggingface ids *dair-ai/emotion*, *rotten_tomatoes*, *sst2*, and *financial_phrasebank*. fMRI data are accessible from https://github.com/HuthLab/deep-fMRI-dataset. PromptSource prompts used as a baseline can be found at https://github.com/bigscience-workshop/promptsource.

## Code availability

Code for running all experiments (as well as applying Aug-imodels in new settings) is available at github.com/microsoft/augmented-interpretable-models and on Zenodo at https://zenodo.org/records/10118975. Code uses python 3.8 and huggingface datasets 2.12.0, huggingface transformers 4.29.0[88–100], sklearn 1.2.0[40], and OpenAI API text-davinci-003.

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

## Author contributions

C.S. and A.A. additionally carried out the experiments and analysis. C.S., A.A., R.C., and J.G. participated in the development of ideas, reviewing of results, and writing and editing the manuscript.

## Competing interests

The authors declare no competing interests.
