## [Peer Review File · Nature Communications]

REVIEWERS' COMMENTS

Reviewer #1 (Remarks to the Author):

The manuscript proposes a method to leverage the information encoded by Large Language Models embeddings and use it in an explainable way, avoiding the black box nature of Large scale neural networks while maintaining a good performance on all benchmarks tested on. The method proposed does so by computing a static (non contextual) version of the LLM embeddings and fitting a linear model (or a classification tree) on the train set after each sample has been encoded in this way.

The manuscript is well written and clear, the experimental setup is sufficient to test the method proposed.

My previous comments on an earlier version of the manuscript have been addressed, and I think the manuscript can be accepted for publication.

I think Table A2 is missing the header which indicates the tasks the scores are measured for and I have a small extra question on the work, now that the additional experiments with different LLM have been added, I wonder if the authors have an explanation for why all other models considered, even those generally outperforming bert do not do so in most of the experiments studied. I believe that the number of tokens used by the tokenizer (or the tokenization strategy) used in pretraining are relevant to this aspect.

More in general if there are interesting relations between the aug-linear tokenizers (NLTK/SPACY) and those used to pretrain the models. It looks like a larger (pretraining) tokenizer might on average influence the performance of aug-linear models more than the size of the pretrained LLM used to create embeddings.

Reviewer #1 (Remarks to the Author):

The manuscript proposes a method to leverage the information encoded by Large Language Models embeddings and use it in an explainable way, avoiding the black box nature of Large scale neural networks while maintaining a good performance on all benchmarks tested on. The method proposed does so by computing a static (non contextual) version of the LLM embeddings and fitting a linear model (or a classification tree) on the train set after each sample has been encoded in this way.

The manuscript is well written and clear, the experimental setup is sufficient to test the method proposed.

My previous comments on an earlier version of the manuscript have been addressed, and I think the manuscript can be accepted for publication.

I think Table A2 is missing the header which indicates the tasks the scores are measured for and I have a small extra question on the work, now that the additional experiments with different LLM have been added, I wonder if the authors have an explanation for why all other models considered, even those generally outperforming bert do not do so in most of the experiments studied. I believe that the number of tokens used by the tokenizer (or the tokenization strategy) used in pretraining are relevant to this aspect.

More in general if there are interesting relations between the aug-linear tokenizers (NLTK/SPACY) and those used to pretrain the models. It looks like a larger (pretraining) tokenizer might on average influence the performance of aug-linear models more than the size of the pretrained LLM used to create embeddings.

Response

We thank the reviewer for their followup comments and are glad that they recommend acceptance. We have updated the header for Table A2 with the appropriate tasks and updated the table caption to reflect the reviewer's point that larger models do not necessarily improve performance. We leave the investigation of tokenizers to future work, as we feel Table S3 shows that Aug-Linear models are fairly insensitive to the two tokenizers we try here (NLTK & SPACY).